# Contributing to evidence-based veterinary medicine: A qualitative study of veterinary professionals' views and experiences of client-owned companion animal research

**Tamzin Furtado**[1], **Elizabeth Perkins**[2], **Debra C. Archer**[1]*

**1** School of Veterinary Science, Institute of Infection, Veterinary and Ecological Sciences, University of Liverpool, Liverpool, United Kingdom, **2** Department of Primary Care and Mental Health, Institute of Population Health, University of Liverpool, Liverpool, United Kingdom

* darcher@liverpool.ac.uk

## Abstract

Research on the outcomes of veterinary treatments in dogs, cats and horses has important benefits for these animals and their owners. However, this information is not always available, and the evidence base is often lower-quality than in human medicine. To identify ways to improve the generation of evidence, we investigated the views of veterinary professionals about research involving companion animal patients and their owners. This qualitative study used semi-structured interviews with veterinary surgeons and registered veterinary nurses working in UK companion animal clinical practice. Interviews were conducted with 20 veterinary professionals from different clinical settings including both those with and without previous experience of research involving animals. Analyses revealed that veterinary professionals valued evidence-based information to help them make informed decisions about treatment with owners. However, there was often not enough available evidence. Veterinary professionals were willing to help produce this research evidence. However, lack of time and resources were key challenges and in addition, they did not always feel they had the necessary research skills, experience and support. Some participants also found it difficult to discuss participation in research with owners of their veterinary patients. They also had concerns about the amount and type of extra information they would need to give to owners. Veterinary professionals also faced a dilemma as their key role is to protect the welfare of animals that they treat, yet felt that there was the potential for some animals assigned to receive a specific treatment as part of a research study to be disadvantaged. Companion animal research has important benefits for veterinary patients, their owners and for veterinary professionals. Based on our findings, more funding, improved research training, resources, support networks and changes to current regulations are needed. Improved evidence would assist veterinary professionals and owners when making informed decisions around veterinary care.

**Data availability statement:** Data cannot be shared publicly because this may identify individuals and they have not given consent for this. Data are available from the University of Liverpool Veterinary Ethics Committee (contact via ivesethics@liverpool.ac.uk) for researchers who meet the criteria for access to confidential data.

**Funding:** This work was funded by the University of Liverpool Policy Support Fund, which is funded by Research England. The funders played no part in the study design, data collection and analysis, decision to publish, or preparation of the manuscript.

**Competing interests:** The authors have declared that no competing interests exist.

## Introduction

Evidence-based veterinary medicine (EBVM) began to emerge as a concept within the veterinary literature in the late 1990's [1–4], following the new paradigm of Evidence-Based Medicine (EBM) that had emerged within human medical practice a decade earlier [5]. EBVM applies EBM principles, classically defined as "the conscientious, explicit and judicious use of current best evidence in making decisions about the care of individual patients" [6], within a veterinary professional context considering veterinary patient needs and client preferences within the process of making a clinical decision [7,8]. EBVM has been promoted globally within the veterinary profession, is incorporated within undergraduate and postgraduate training programmes and is embedded within professional regulatory requirements. Concurrently, resources and networks have been developed including evidence syntheses, evidence-based clinical guidelines and improved access to published research [9].

In human healthcare, EBM is promoted and facilitated through a variety of professional and academic networks, e.g., British Medical Journal Best Practice support tools [10]. Regularly updated, high quality, accessible synthesised evidence generated by organisations such as the Cochrane Collaboration are also available to address important, current questions around health and care decision-making [11]. This requires availability of high quality, unbiased, primary evidence involving engagement of medical and other healthcare professionals and patients with research studies. These are underpinned by well-established ethics frameworks governing studies involving human participants. However, concerns around clinical research involving vulnerable patient populations, such as children, who cannot usually consent for themselves adds to the complexity of undertaking patient-based research, potentially excluding those populations from clinical research and evidence production, and the associated benefits of EBM [12].

The latter complexities have important parallels with research involving veterinary patients where owners of animals are required to provide consent on their behalf. Previously identified barriers to the use of EBVM within veterinary clinical practice [13–15] include a lack of available high quality primary evidence based on research conducted in veterinary patients in clinical settings. However, exclusion of veterinary clinical patients from high-quality primary research also excludes this population from the benefits of EBVM. Whilst research involving experimental laboratory animals is tightly regulated and relatively consistent across countries, currently, there is no internationally agreed framework for the involvement of animals in receipt of veterinary care with regard to their participation in research [16]. Ethical regulations are set at a national or regional level and with variations in ethics, regulations and professional body frameworks between countries. Within the UK, veterinary clinical research (VCR) is defined by the Royal College of Veterinary Surgeons (RCVS) as routine procedures undertaken for the benefit of the animal/s, with the concurrent intention to generate new knowledge that benefits animals and requires ethical approval [17]. This is different to the RCVS definition of routine veterinary practice (RVP) where procedures and techniques are performed on animals by registered veterinary surgeons (or registered veterinary nurses under their direction) in the

course of their normal professional duties (i.e., not for the purpose of generating new knowledge). Where a procedure or treatment is not considered to be RVP, additional regulatory approval is required [17].

Medical professionals are frequently involved with patient-based research with increased calls for research to become normalised as core business within the National Health Service (NHS) in the UK [18]. However, veterinary patient-based research is less common despite its importance to veterinary professionals and animal owners alike [19]. Identification of factors that motivate (facilitate) or act as barriers to engagement of veterinary professionals with EBVM has been the subject of other studies [15]. However, limited research has been conducted around the role of veterinary professionals in generation of primary evidence as part of EBVM. The aim of this qualitative study was to explore UK veterinary professionals' views around their own contribution to primary evidence generation through research involving client-owned companion animal veterinary patients (dogs, cats, and horses), and their assessment and application of existing research to inform clinical decision-making.

## Methods

### Ethics statement

The study was approved by the University of Liverpool Veterinary Research Ethics Committee (VREC1193). All participants gave written or audio-recorded informed consent prior to participating. Recruitment of participants took place between 02 June 2022–23 February 2023.

### Study design

This qualitative study used interviews of UK-based veterinary professionals (veterinary surgeons [veterinarians] and registered veterinary nurses) to provide in-depth exploration of participant's views and experiences of research involving client-owned companion animals (defined for the purpose of this study as dogs, cats and horses) and contribution to, and use of, EBVM.

Key areas that we sought to explore included: veterinary professionals' prior experience of, and willingness to participate in, patient-based companion animal research including interventional studies such as clinical trials; opinions around commercial (pharmaceutical industry) vs academic (charity or veterinary sector funded) research; practical challenges in recruitment of patients and study conduct including data collection. A semi-structured interview script (S1 Appendix Semi-structured interview guide) and vignettes of 3 clinical studies (Table 1) were developed and refined following feedback from veterinary professional colleagues. The vignettes placed the research activity in different types of research context: an academic research study investigating two medical therapies; surgical procedures considered to be routine veterinary practice (RVP according to RCVS guidelines) but to which a patient would be randomly allocated to (therefore defined as clinical veterinary research [CVR]); a commercial (pharmaceutical industry funded) clinical trial investigating a novel therapy in client-owned veterinary patients.

**Participant recruitment.** Veterinary professionals (veterinary surgeons and registered veterinary nurses) working in, or with experience of, UK companion animal (small animal and equine) clinical practice and who were willing to be interviewed were recruited. Participants were initially recruited through advertisements on veterinary professional social media groups and platforms, and by snowball sampling. Initially, any participant who applied to take part was accepted in the study. As recruitment progressed, sampling was strategic to ensure a diversity of participants across the sector and including those without and with different types of prior research experience.

**Interviews.** Semi-structured interviews were conducted by TF (non-veterinary professional) using video-conferencing software (Zoom.com). Interviews were based upon the semi-structured interview script already described but were adapted to the individual's experiences. Video interviews were audio-recorded and professionally transcribed. Any identifiable data, such as the names of participants, veterinary practices, or involvement in specific research studies were anonymised, and the cleaned transcripts were then uploaded into NVivo (2020) for analysis.

**Table 1. Vignettes used during the interviews to stimulate discussion around a semi-structured interview guide.**

| Scenario | Study descriptor | Small animal example | Equine example |
|---|---|---|---|
| 1 | Randomised study comparing two routinely used medications | Your practice has been asked to take part in a study comparing two routinely used medications (Medication A & Medication B) used as painkillers following orthopaedic surgery in dogs. Dogs would be randomly allocated to receive either Medication A or Medication B. You (or the vets in your practice) usually use medication B. You (or the vets in your practice) usually use medication A and think it works well but have not used medication B. | Your practice has been asked to take part in a study comparing two routinely used medications (Medication A & Medication B) used to medicate joints for management of arthritis. Horses would be randomly allocated to receive either Medication A or Medication B. You (or the vets in your practice) usually use medication A and think it works well but have not used medication B. |
| 2 | Randomised study comparing two standard surgical techniques | Your practice has been asked to participate in a study comparing two commonly used surgical techniques for performing a bitch spay [ovariohysterectomy] (Technique A and modified Technique B). Dogs would be randomly allocated to undergo either Technique A or Technique B. Within the practice vets perform both techniques but some vets prefer Technique A and others Technique B | Your practice has been asked to participate in a study comparing two commonly used surgical techniques for performing equine castration (Technique A and modified Technique B). Horses would be randomly allocated to undergo either Technique A or Technique B. Within the practice vets perform both techniques but some vets prefer technique A and others technique B. |
| 3 | Commercial drug trial | Your practice is asked to participate in a commercial drug trial looking at a new drug in cats diagnosed with Feline Infectious Peritonitis (FIP). The product has been used in another country and the company are looking to get it licenced in the UK by conducting a clinical trial assessing safety and effectiveness of the drug. All drug and associated costs are covered by the drug company. | Your practice is asked to participate in a commercial drug trial looking at a new injectable medication to treat Cushing's disease (Pituitary Pars Intermedia Dysfunction; PPID). The product has been used in another country and the company are looking to get it licenced in the UK by conducting a clinical trial assessing safety and effectiveness of the drug. All drug and associated costs are covered by the drug company. |

**Analysis.** Analysis was based on a constructivist Grounded Theory as described by Charmaz [20], with a social constructionist epistemology [21]. Data were collected and analysed concurrently, meaning that initial analysis informed later interviewing and analysis. Initially, open-coding was performed, with TF reading each transcript and writing down impressions and initial thoughts. Secondly, more targeted coding was used to 'label' units of text which were of particular interest; for example, text that showed assumptions, values, actions, or language use. As more data were combined into the dataset, codes were continually refined, clarified, and re-named in order to incorporate the information within them, and to represent the ideas being expressed by participants. 'Memos', or notes regarding broader interpretation, were kept on a virtual whiteboard throughout this process. As the research progressed, the memos and codes were organised to create overriding themes which allowed the researcher to understand the views and experiences shared by interviewees.

TF also used memos to reflect on her positionality of the study; for example, her own prior experience as a participant in in several clinical research studies, decision-making and evidence-seeking around maintaining health of her own companion animals, and reflection on prior discussions with veterinary colleagues around research. In line with the grounded theory approach, in the initial phases of analysis, no particular lens or theory was applied so as not to limit or impact the coding process. As the analysis phases became more theoretical towards the very end of the process, existing theories and approaches which offered additional perspectives on the data were considered by TF in conjunction with EP and DA. At this later point of analysis, Zinn's body of work on perspectives on risk was identified as reflecting issues discussed by participants with regards to their clinical decision making, and thus provided a theoretical lens which added further insight to the analysis.

## Results

### Participants

A total of 20 participants including 15 veterinary surgeons (VS) and 5 registered veterinary nurses (RVN) were recruited from across the companion animal veterinary sector. In-depth interviews were conducted online for each except one participant who chose to provide detailed written answers to the questions via email. Study participant details including species, veterinary professional role and veterinary sector worked in are shown in Table 2. To preserve participant anonymity, details of participants' prior research experience are not individually reported. A total of 5 participants had no prior research experience (3 = VS, 2 = RVN), 4 had laboratory animal research experience but no clinical research experience (1 = VS), had or were currently undertaking a formal postgraduate research (PhD) qualification but had no clinical research experience (3 = VS), 8 (6 = VS, 2 = RVN) had some experience of clinical research (e.g., as a collaborator or principal investigator or an observational study) and 3 were experienced in clinical research and conduct of interventional studies either as a research nurse (1 = RVN) or principal investigator (2 = VS).

### Interview results

In this study, the veterinary professionals interviewed described working in a culture which they perceived to value ongoing learning and a sense of "professional curiosity". Paradoxically, however, the practicalities of working as a veterinary professional hampered their ability to keep up to date with new evidence, or participate in evidence generation. Use of EBVM principles was viewed by participants as a fundamental part of their veterinary practice, involving questioning their own current practice and seeking new information to provide optimal treatment for the benefit of their veterinary patients and owners. Clinic-based research was perceived as being integrally tied in with EBVM given that it involves questioning dogma and producing new evidence for continually improved practice. However, the structure of the veterinary profession was viewed as problematic by study participants. Veterinary professionals described feeling overworked and lacking time to undertake key clinical tasks, making it difficult to actively engage in other activities such as research. The structure was at odds with the values of the veterinary culture, because it promoted "status quo" treatments and interventions,

**Table 2. Features of veterinary surgeons (veterinarians) and registered veterinary nurses who participated in interviews.**

| Companion animal species worked with | Veterinary professional type | Veterinary sector worked in |
| --- | --- | --- |
| Small animal | Head veterinary surgeon | Primary care – University teaching practice |
| Small animal | Veterinary surgeon | Primary care – University teaching practice |
| Small animal | Veterinary surgeon | Primary care – private veterinary practice |
| Small animal | Veterinary surgeon | Referral hospital - private |
| Equine | Registered veterinary nurse | Equine hospital (primary care and referral) |
| Small animal | Registered veterinary nurse | Research nurse - Referral hospital |
| Small animal | Veterinary surgeon | Primary care – private veterinary practice |
| Small animal | Registered veterinary nurse | Primary care – private veterinary practice |
| Small animal | Registered veterinary nurse | Research nurse – University referral hospital |
| Small animal | Veterinary surgeon | Referral hospital & research - University |
| Equine | Veterinary surgeon | Self-employed |
| Small animal | Veterinary surgeon | Primary care – private veterinary practice |
| Equine | Veterinary surgeon | Primary care – private practice |
| Small animal/mixed practice | Veterinary surgeon (practice partner) | Primary care – private practice |
| Small animal | Veterinary surgeon | Referral hospital |
| Equine | Veterinary surgeon | Referral hospital - University |
| Equine | Veterinary surgeon | Pharmaceutical company |
| Equine/ mixed practice | Veterinary surgeon | Primary care – private practice |
| Equine | Registered veterinary nurse | Primary care – private practice |
| Small animal | Veterinary surgeon | Primary care |

hampering participants' ability to be curious and engage in ongoing learning or evidence generation. In Fig 1 the interplay between the individual, culture, structure and EBVM is illustrated.

***Contributing to the evidence base: veterinary professionals' views on client-owned companion animal research:*** Most participants were familiar with the concept of CVR and held a favourable view of running research studies within a clinical setting. All participants agreed that pragmatic research questions which answered every day clinical scenarios should be prioritised, and that patient-based research would provide genuinely useful, robust primary evidence. Engaging in research studies and new evidence synthesis was compatible with a sense of "*professional curiosity*"(P13), in which veterinary professionals reflected on their practice and sought new information. Research was seen as potentially adding to the collective evidence base for day-to-day veterinary practice:

P2: *At the end of the day vets are scientists.*

P15: *I think if we are to make progress and not just reliving Groundhog Day style the same mistakes of all our predecessors, we are going to need to engage with clinical research, and in an environment where it's relevant.*

Additionally, while in theory participants were positive about taking part in clinic-based studies, the case study scenarios provided insight into some more specific areas of contention. Every participant discussed the need to minimise potential harms to veterinary patients participating in research studies. For some participants these potential risks needed to be reduced so far as to make the experience of an animal enrolled in a research study indistinguishable from an animal undergoing standard veterinary care. For example, comparing two known and licensed analgesics was generally considered ethically permissible. However, studies involving a medication with unknown or limited evidence of efficacy raised concerns. Responses to the scenarios exposed a tension between the veterinary professional acting as a provider of care for the individual animal, and a researcher who was using an animal to generate clinical evidence, and all participants

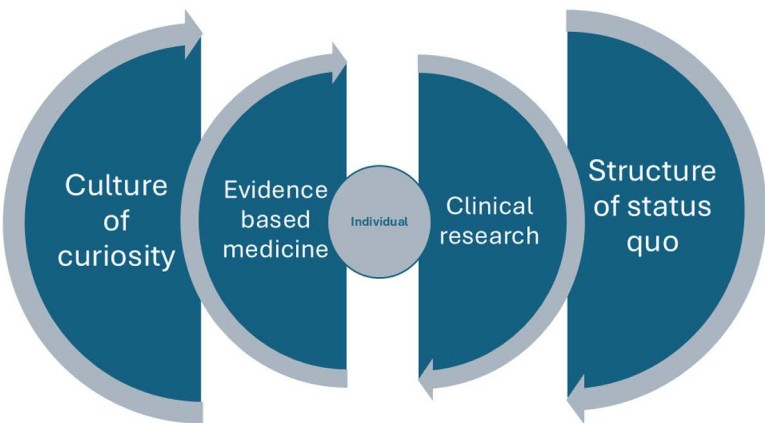

**Fig 1. Model showing the interplay between factors found in this study.**

described "animal welfare" as their first priority. As a result, studies that might expose animals to any level of negative experience were considered problematic and potentially unethical:

P1: *I think if you were doing a drug A versus drug B, I'd need to know that drug B works. I know the whole point of testing is to test which one is superior, but I'd need to know that the drug had pain relief, and was proven for pain relief, on its own. I know it probably would have been, through the licensing process, anyway. But I'd need to know that I'm not, sort of, permitting half my dogs to be in pain.*

P16: *I won't give a placebo if I don't feel that it's in the animal's best interest and I'm a very clinical, clinical researcher. I have always done, I've observed, collated, asked questions and you know performed interventions that I feel are in the animal's best interests and compared with standard treatments. But I've not withheld treatment for the sake of research and I would never do that.*

Placebo-controlled studies were generally considered unethical by participants in this study, due to the need to withhold treatment from the control group. However, randomised trials comparing known interventions were perceived as acceptable because both groups of patients would receive treatment.

Randomised studies were sometimes contentious for participants, because of the need to discuss the lack of evidence-base for an intervention with clients. Some participants described that in clinical circumstances, they would simply give a client a medication or advise a particular intervention. The client would trust that their animal had been given an appropriate treatment, usually without further discussion about the available evidence for the treatment. However, in a research scenario, the lack of evidence for the intervention and control would be described to the client as part of the consent process, creating an unusual client-veterinary surgeon dynamic:

P14: *Most owners, they go home with the dog's painkillers, you know, now they wouldn't know if it was A or it was B. They would just assume you are going to send it home with pain relief. You're then saying to them right I'm going to give them painkiller A. or I'm going to give the dog painkiller B. they are going to say well what's the difference, you know "I don't know".*

This was termed "revealing uncertainty" in the coding. It incorporated ideas around managing animal welfare with the unknowns which were described overtly in the research scenario, yet remained hidden in standard care, with the client

more frequently placing their trust in the expertise of the vet and the recommended treatment. Revealing uncertainty to clients as part of the research procedure flouted the expected norms and trust between professional and client. This unfamiliar dynamic could be perceived to undermine veterinary surgeons' credibility to some extent, and also place an additional burden of decision-making on clients, who may have to weigh up evidence themselves.

However, participants discussed instances within veterinary practice during which uncertainty was present, yet without the same level of concern. For example, newly licensed drugs, off-label usage, or veterinary "specials" (extemporaneous preparations of a product without marketing authority) as part of clinical care did not usually concern participants in the same way as a patient-based research study. In this situation, veterinary surgeons described that they would inform the client that the product was off-label or a "special", but suggested that the depth of discussion was different to a veterinary patient in a research study, in which all aspects around potential uncertainty were required to be explicitly stated to the owner of that animal. In this way, the relationship and dynamic of trust between client and professional was maintained, despite uncertainty.

As a result of the need to reveal uncertainty around interventions in patient-based research and the need to protect animals from potential harm, most participants considered Scenario 1 (a randomised trial on two licensed drugs) to be acceptable only when there was adequate available data to support the efficacy of both drugs. Additionally, a "rescue protocol" (a pre-agreed plan for analgesia if the study drug is not working) was also viewed as necessary by participants with prior experience of conducting these types of studies. Participants had an additional concern about Scenario 2 (a randomised study comparing two routine surgical techniques for a procedure) because of the potential for a veterinary surgeon's lack of familiarity with the procedure to add a layer of risk for the patient:

P9: *I would worry that the vets would have, if one vet was given technique B, but they're used to doing technique A, then I would maybe have a little bit of reservation about how skilled the vet is going to be in doing that technique… Would they then struggle to do it in this new technique that they're not quite confident with, which then could have an impact on the surgery time?*

Contrastingly, Scenario 3 involved a pharmaceutical company funded clinical trial investigating a novel therapy for management of a chronic condition, feline infectious peritonitis (FIP), in the small animal context, and Pituitary Pars Intermedia Dysfunction (PPID) in the equine context. In this instance, respondents were generally comfortable with offering the drug given the few available options for both conditions. In comparison to PPID, FIP is a usually fatal condition with no effective therapies previously available. Hence participants described that they would be comfortable trialling a new medication under this circumstance:

P8: *In the case of FIP we are often stuck as to what we are going to do. So, I would be quite happy in that sense to sort of say yes this is something we could try.*

P2: *FIP essentially, for most cats is a death sentence. So, I think if there's any option of treatment of trialling something that could prolong life or effectively treat it then that's a good thing.*

Across all three scenarios and differing professional and personal experiences, participants frequently imagined discussing informed consent with owners of eligible animals, in order to help them weigh up their own values and ethical views around the research study:

P14: *Back 10, 15, 20 years, FIP was a death sentence so you'd say to an owner "your cat has got FIP. We've got two choices. I can put your cat to sleep now or we can fill it full of some steroids for a few weeks and then we will put it to sleep in a few weeks/months". Alternatively, "here is the new drug, don't worry about the cost because that's all being*

 

*covered, now we can try it but your cat you know might have side effects, might have problems, it might die. But I've just told you your cat is going to die anyway, we can try it, it might work so we've got nothing to lose have we?"*

The prospect of having a discussion around informed consent with an owner acted as a mechanism for participants to evaluate their own ethical views around the research study, in the context of specific client-animal dyads. Examples included taking some responsibility for deciding whether a client and their animal would be suitable, or it was in their interest to be invited to take part. Informed consent was described as a process taken extremely seriously by all participants. Participants described multiple ways that they would ensure that clients maintained autonomy when making decisions around their animal being recruited onto a clinical trial. Examples given included simplifying language, assessing comprehension, asking the question about research participation at an appropriate time, and being honest about the potential risks and benefits of participating:

P19: *If I am asking the client to consent, I try to keep my opinions out of it, and just give them the facts and let them kind of decide unless they asked my advice on it… if I was unhappy with the level of understanding with the owner, and I didn't think that they could do informed consent, then I would be reluctant.*

P1: *The only thing would be making sure that I'm not accidentally giving them false hope.*

Participation in studies place responsibility on the owner to make an ethical decision on behalf of their animal. The veterinary professional is required to convey often complex concepts to the client to ensure that this decision can be made. However, participants described situations where clients had not listened during the informed consent process for standard clinical interventions, such as companion animals undergoing routine neutering. This made participants question whether animal-owners could play their own part in the decision-making process effectively:

P14: *The problem [with Scenario 2] is you know an owner is coming in, they are overwhelmed with all the information about the big technical surgery I am going to do today anyway.*

**Barriers to participation in patient-based research:** Prior to taking part in research studies, several participants described feeling that research was interesting yet intimidating potential component of veterinary work, perceived as something done by people with specific attributes, skills and knowledge:

P4: *I think a mental barrier at least in my head probably until working alongside people like, realising you didn't have to be super smart and academic to do that kind of stuff [research], actually it was a range of people.*

P17: *I always thought it would be really interesting to do research in clinical practice and actually it shouldn't be that hard to do. You should be able to do it, it should be fine, but knowing where to start was always off-putting and I didn't know where to take the first step, how to do it.*

Often "knowing where to start" was a significant barrier; for example, knowing how to design a study, find out about licenses required or how to submit an ethics application. This could be frustrating for some. For example, P13 had research questions and had stored clinical samples as part of their routine work outside the UK, with the knowledge that they would be interesting to research. However, they did not have the knowledge, time or funds to take their research question further and had been unable to source collaborators:

P13: *It [research] certainly is something that interests me. It's just very, very difficult to know how to get into it. As regards research I do very, very much have a professional curiosity about things and so do my colleagues, but how that translates into research is kind of difficult. We often don't know, we've some really good data and we often don't know how to use it.*

Other participants, too, described the difficulty in seeking collaborators who would help them to get involved in clinical research:

P17: *I wish there was an easier forum that would probably have to go through a university of people being able to sign up to clinical research in practice. So, Professor whoever is running this study, do you want to be involved in it and having a way you can go and look at that and get involved, because necessarily coming up with the questions is quite hard. Coming up with that initial stage is quite hard, but I think if it was a case of presenting it to a client, getting a form filled out and then reporting the data, I think there would be a lot of vets who would be interested in doing that.*

Conversely, however, participants who were confident leading patient-based research studies reported finding it difficult to engage veterinary professionals to take part in them. P15, for example, had prepared a research protocol with a pragmatic research question which they felt could be easily answered by performing a low-risk study in collaboration with other veterinary practices. They described how they worked with veterinary professionals within those clinics to ensure that they were an integral part of the research, designing easy to use data collection materials, and providing ongoing support. However, they found that clinical staff did not enrol animals into the study due partly to time constraints and the pressures of general practice:

P15: [at the beginning] *they were brilliant, they were super supportive and there was a massive enthusiasm to get involved in this. All the centres took home all the kit and stuff and very, very positive feedback initially. But again, I think that people are very busy, there are lots of other stresses and they find excuses, or good reasons I'm sure not to enrol any particular patient and after a year you ask them how they're doing and they say "well not brilliantly".*

As demonstrated in P15's example, a lack of time or concern about disrupting fast-paced veterinary clinical practice was the most-often cited reason for not participating in clinic-based research. This was followed by concern by some participants about recruiting eligible animals due to a lack of confidence in the ability or willingness of the owner of that animal to provide consent. Additionally, participants described feeling that clinic-based research was onerous and complex, and might also mean less time spent with their patients due to the need to spend more time on paperwork. Veterinary professionals who had experience of running patient-based research studies in clinical settings described the need to work collaboratively with veterinary professionals within the research team and with animal owners. To overcome these barriers, they increased confidence and a sense of ownership of the research project:

P16: *If you come to your team of residents, interns and staff and say "oh I've got this funding aren't I clever", you are just going to put their back up, and they are just going to see this as extra work… [in a past project] I had an administrator at the time,… I said to her "are you interested in helping me with you know looking after these cases?". She loved collating all the information and she worked as research assistant part time for me and off we went, and again working together as a team with a problem, going together to get the ethics, going together to get the funding that's how you bring in the team, you don't turn up with a project.*

As P16 reported, the manner in which a research project was presented influenced the willingness of others to take part in patient-based research tasks outwith their daily professional responsibilities. However, as noted by P15, results were mixed, thus hampering their own efforts to perform ongoing research studies.

***Stepping stones to engaging veterinary professionals with patient-based research:*** Several participants described how they used clinical audit as a means of getting other veterinary professionals to take an initial step into research. Clinical audit was viewed by some participants as a cross between clinical practice and research. This produced meaningful results, but without many of the methodological components such as research ethics approval, informed

consent, or sample collection. Participants who became involved with research either did so by proxy (for example, help-ing take samples for a study being run by others), or through post-graduate training such as undertaking a professional Certificate, Master's degree or doctoral training. This helped to build their confidence around their own research skills in a situation where time had been specifically allocated for this purpose.

Once engaged in research studies, participants reported a range of benefits, from improved team-working through to learning new skills, and seeing additional career opportunities. This was particularly emphasised by registered veterinary nurses, who found a new niche available to them; two of the participants were employed specifically to work on research studies:

P9: *The career opportunities are massive … I don't think I realised when I was in first opinion practice, I thought it was kind of like veterinary nursing in a first opinion practice, in a referral centre, or maybe a rep and that was it. But actually, there is so much more that vet nurses can do lecturing, teaching, research, there's just so much more that I just didn't realise before I started this really.*

Participants who had prior experience of undertaking patient-based research also described how taking part in studies potentially offered enhanced opportunities to spend time with animal patients and their owners during study visits. How-ever, those who had no prior research experience were concerned that doing so might limit their time spent with other non-participating clients and animals, due to demands of time required to complete the required research paperwork.

Participants also described the importance of making the research study as pleasant as possible for the veterinary pro-fessional, animal and its owner(s). This involved making study procedures as straightforward as possible and making sure the client felt that their contribution to the study and collective veterinary knowledge was appreciated. Generally, clients were said to respond favourably to being asked whether they might be willing to enrol their animal in a research study, even when the study requirements were quite onerous.

P4: *It was to our advantage that we were offering, at least with that study, a very high standard of veterinary care... it probably was about £200 worth of veterinary care a year which, especially for a lot of the people in that socioeconomic sort of demographic and the part of [place] were the clinic is, that was a really big incentive, they loved to be able to provide something like that for their cat.*

P10: *When I've explained, well, actually, if your dog participates now, when we publish all our findings, when this is out into practices, your dogs have had the opportunity to try and prevent other dogs from going years and years and years with hav-ing a lameness that hasn't been detected…. And that's I think, when they've been able to see the benefit of the research, in later years for dogs in the future. That's what's really sparked their interest. And they've all been really, really enthusiastic.*

Participants with experience of conducting veterinary clinical trials also described ways to potentially reduce some of the concerns that less-experienced researchers may have. This included the protocol for withdrawing clients' animals from the study or using a rescue medication if needed.

P5: *I know that within our randomised trials if, at any point, the owner of the horse has any concerns that horse will be removed and then you can unblind that case and you can then continue treatment accordingly. So, I guess the option is always there to take a step back and to take yourself out of that situation if you were not happy with it…. I equally don't feel that anybody within our industry would do something they felt would be detrimental to the horse.*

In this way, concerns about maintaining optimal animal welfare during a patient-based research study were overcome, clients had reassurance of their autonomy over their animal's ongoing inclusion, and it provided reassurance to veterinary professionals who had concerns around potential ethical issues.

***A culture of curiosity: use of evidence-based veterinary medicine (EBVM):*** EBVM was constructed as a central component of veterinary professional culture, comprising decision-making based on evidence from clinically relevant research studies. Participants described how they strived to maintain an up-to-date knowledge of evolving research findings. Use of skills in critical thinking to apply this knowledge to veterinary clinical practice was thus constructed as a central tenet of being a "good" veterinary professional. As part of this willingness to learn and update their own professional practice, participants were encouraged to question dogma and traditional practices:

*P16: I think there's a real push for every vet in practice to question even the basics, you know, should I be disinfecting the skin before I give an IV injection or is it better to be dry. You know, should I? And sometimes the answer is no and sometimes it not.*

*P13: Your mind is always thinking and turning over and you're asking yourself why or how, how could we do better. Should we be doing better? Are we making any difference or is it just time? Are we just going to do the same thing over and over and expect a different result?*

*P12: So much of what we've done has been from, you know, James Herriot days and its only now that some of these things have been questioned. For instance, we used to standardly neuter everything that walked through the door and now it's actually, we are questioning that.*

The culture of the veterinary profession was described as valuing this mindset and a "*passion*" for continual improvement in patient care. A 'good' veterinary professional was viewed as being able to maximise health for animals in their care by being aware of the efficacy of possible treatment options and the potential for adverse effects. This enabled them to provide an informed presentation of potential options to animal owners, often to the point of being able to quantify risk for clients:

*P5: I see it every day, when people ring, particularly in the middle of a colic surgery and an owner says "right well what's the prognosis can you help me make my decision?" and actually I say the majority of time that that information in our environment comes from research and we are being able to put a quantitative figure on something to help people make a decision.*

Regular revisions of practice protocol due to evolving evidence were expected. Participants found it relatively easy to think of recent examples when the veterinary practices and hospitals they worked within had made changes based on new clinical evidence, for example, when practices altered their protocols around antimicrobial use. Individuals who did not show enthusiasm for keeping up to date were described as "*dinosaurs*" (P12), and considered to have "*lost their passion*" (P7), and were considered to be potentially poor veterinary surgeons because of their lack of awareness of veterinary advancements:

*P1: I've definitely locumed at practices and worked at practices where I've not been senior, and it's been on the opinion of the lead vet. And if they're someone who's very….traditional, stuck in their ways – I hate to say it, but old school boomer vets – it can be difficult to say "we should be doing X, Y, Z" because "Oh we've always done it this way, it's always been fine for me". It is then very personality dependent, but a lot of practices I've locumed at that's been the situation, I've then been offered permanent jobs and turned them down because I've sort of, self-selected to be in the kind of teams I like to be in, which is open to new stuff, happy to change things around.*

Participants described how they saw it as part of their job to take personal responsibility for keeping up to date with new research findings. This involved a range of methods including reading and critiquing research articles, reading

news, discussing cases with colleagues, studying for postgraduate qualifications, attending conferences, seminars and continued professional development (CPD) events, and in-practice sessions organised by pharmaceutical companies, often referred to as "lunch and learn". These activities could be active (sought out by an interested individual) or passive (attending only mandatory CPD, or only in known areas of interest; going to lunch and learn sessions because they were convenient).

A passion for practising EBVM and a positive attitude to clinic-based research were considered to be integral to each other. The mindset required to be a good veterinary surgeon was constructed in the same way as patient-based research, given both required the need to make use of a professionally curious mindset:

> P1: *It's a funny one, trying to make people realise they do do research every day, it's just maybe not how they think of it. Not big drug trials with fancy professors, it's "look our cat castrates are looking poor in recovery, let's look at their pain scores and see if we can change something".*

> P16: *You asked me about research versus practice, but I think its categorising it as two different things. I think they're not two different things, I don't think you can be a good practitioner without asking questions.*

***A structure of maintaining the status quo: difficulties practising EBVM:*** Despite the culture of ongoing learning, evaluating evidence and keeping up to date, this was not necessarily considered easy for participants and was attributed to structural issues within the veterinary industry. These predominantly related to the poor availability of an appropriate evidence-base for making evidence-informed decisions, such as inability to access journal articles, and limitations on professionals' time and energy. This made it difficult for them to keep up to date with new information, or think about participating in clinical research. Combined, these issues contributed to participants' reliance on their previous experience and a "*paint by numbers*" (p15) approach to treating clients' animals, rather than maintaining a professionally curious mindset and conducting ongoing learning.

The context within which participants worked was textured by a shortage of veterinary professionals, a highly demanding workload, and clients who had increased expectations with a willingness to resort to litigation to settle disputes. The task of keeping up to date with new research took a backseat behind the more immediate concerns of managing the day-to-day work context:

> P12: *We are all stretched. And like my day yesterday was just horrific, constant, and sometimes you just don't feel like you're doing a good enough job for your patients and not caring for them as well as you could do just because you're literally being pulled from pillar to post and being hit like a dodgeball everywhere.*

> P2*: I don't recall ever being at a practice where we routinely had journal clubs. It was attempted to be set up at the university practice when I first started there and it did last a little bit but unfortunately it was just, you're so time pressured, it wasn't able to be fitted in in the day. We tried to do it on the evening, but inevitably, by the time you finish at work in the practice its later anyway and then people want to go home.*

> P17*: I think what lots of people would say, if you asked them why, what are your barriers to interpreting the evidence, they would say time. I don't have time to go and read papers, I don't have time to evaluate it, I don't have time to sit and think about it, I don't have time to drink a cup of tea!*

Additionally, most participants described feeling that, even with their best intention to practice EBVM, there was a lack of clinical evidence on which they could base their practice. Examples given included the frequency of veterinary research studies that had small sample sizes and thus low statistical power, or because studies were not considered applicable to a general practice setting:

*P7: For most drug trials, the study numbers are poor aren't they…. most maybe like 40, 50 patients in a trial and although, you know, maybe the P Value says that it is statistically relevant it still doesn't feel sometimes like you can completely just trust that, I don't know.*

P16: *You only have to look at Veterinary Evidence, the journal which looks at knowledge summaries and you know more than half of them go well there is not enough evidence to answer this question, there is just not enough evidence. A lot of the evidence is poor quality.*

P15: *I would love to work out those doors into* [conducting research in] *primary practice, because ivory tower work done in universities doesn't always apply to the standard treatment.*

In addition, the structure of funding for patient-based research and research publication was viewed as potentially heavily biased. Participants reported being unable to trust that published information provided an honest reflection of the results of research studies, particularly if those results were generated or funded by organisations with a vested interest in the results, such as a pharmaceutical company. Keeping up to date with new studies therefore required not only reading the available evidence, but critically appraising the methods and funding models. Some respondents described that establishing the "truth" of a treatment was complex, time-consuming, and sometimes disempowering:

P14: *I think there's a lot of information but trying to then narrow that down, or trying to filter it out, I think that often is the difficulty, you know that you can, especially with something as grey and non-black and white as medicine and biology, there can almost be, for every argument for something you can almost find something that's against it, for me to know in general practice if a particular study is flawed or not, or how good that study was, you know on something that I'm just reading online - I am not skilled enough to know how, 'cos there are studies, as you well know, there are studies and there are studies aren't there?*

Contrastingly, many participants described trust in the licensing process, thus being reassured that a licensed product was appropriately safe and efficacious for use in clinical practice:

P12: *From an ethical point of view having done the work I've done, I know that in order to get the licenses and permissions and the ethics approval then it's gone through the various hoops to get through that.*

P19: *As long as it's proper authorised veterinary medicine, and proper trials I wouldn't be too concerned. But there's always a little bit of concern when you use a new thing.*

Licensing was one example of many, in which where participants described relying on trust (whether in a process, a brand name, or a type of industry, e.g., academic research vs pharmaceutical), tradition, or experience to allow them to function in the busy and stressful veterinary companion animal clinical environment.

## Discussion

While veterinary professionals prize EBVM and view professional curiosity around veterinary medicine as part of being a "good vet", they reported that working as a veterinary professional meant that they had a lack of knowledge and time to read about or take part in research. To our knowledge, this is the first study to investigate veterinary cultures around knowledge production and application in client-owned companion animal patients. Our findings have identified potential approaches to improve generation of primary evidence and promote engagement of veterinary professionals in patient-based research.

The acceptance of EBVM principles and aspiration to utilise evidence-based practice as a responsible, modern veterinary professional is not just a "passion" as described by some participants, but an epistemological approach and belief system with roots in medical literature which prioritises the seeking of "truths" via science [22]. Participants described that

veterinary professionals should have a sense of 'professional curiosity' in using knowledge produced through scientific research to try to quantify uncertainty in the management of clinical conditions or determine whether equipoise exists. However, they described that they were constrained by the pressures of service provision, and current pressures in the veterinary field pertaining to the cost of treatments, transparency of available options, and clear information around treatment outcomes [23]. Clinical research has an important role to play here in clarifying treatment efficacy and increasing the treatment options available, by adding to the knowledge base of clinical evidence. This information aligns with Zinn's "rational" approaches to risk, in which positivist cognitive approaches to decision-making are viewed as more effective than "irrational" approaches, which include hope or faith, or "in-between strategies", such as trust or intuition [24].

However, participants also understood that the evidence generated through veterinary patient-based research was constructed through complex, socially constructed processes where certainty was not always achievable [25]. Borrowing from Zinn's work applied to the veterinary clinical setting, participants valued research that armed them with logical answers to clinical questions, such as the best treatment option for a condition. Whilst they presented themselves as predominantly "rational" thinkers, the demands and structure of veterinary clinical practice resulted in numerous examples of "in-between" approaches to risk at play [24,26]. Practice was constructed as a colourful, fraught, messy reality of interaction between professional, client, and animal where rational approaches were problematic and, sometimes, impossible [27].

Webs of trust and other "in-between" approaches to navigating clinical evidence and decision making were also identified in our data. Participants described the tensions around trusting clients to listen and understand options for treatment [28], sought science-based answers to clinical questions yet used intuition as part of "clinical skill" [29]; discussed trust in drug licensing and production procedures, and described emotions such as fear when trialling a new product. The uncertainty presented when conducting randomised studies in particular added additional layers of complexity. Participants overtly discussed moral tensions around evidence generation, including clinical equipoise and use of placebos [30,31]. Communication and trust between clients and veterinary professionals during clinical research has been researched, with a focus on informed consent for clinical procedures and protecting veterinary professionals from alleged professional negligence [27,28,32,33]. The extensive role of trust between client-professional communication identified in this study is important when considering veterinary professional perspectives on patient-based research.

Implicit uncertainties in practice are made overt during the research and informed consent process, by laying bare the holes in clinical knowledge which are often overlooked during veterinary consults [28,34]. Initiation of discussion about a patient's potential participation in a research study and obtaining informed consent was therefore an important obstacle for some participants in the present study. This was further compounded by concerns around the additional paperwork and complexity of information [35] that veterinary professionals might be required to give to a client about a research study, particularly within a time-constrained clinical environment.

Based on the findings from the present study, there is a need for further research and training around communication between veterinary professionals and clients in the context of companion animal patient-based research. The development of evidence-based frameworks, similar to those used in medical research involving paediatric patients [36], could enhance veterinary professional explanation and client understanding, together with improved resources and networks on this specific aspect. Researchers and ethics committees must ensure that consent forms and associated information meet existing evidence-based healthcare readability guidelines [35,37]. Engagement with owners of companion animals during the design of veterinary clinical research studies has important benefits [34]. Their involvement during the design of client information and consent forms may help to address concerns expressed by participants in the present study around the type of information presented to clients together with initiation of discussions and type of language used. This process is important to ensure the process of informed consent has been met and to optimise engagement of veterinary professionals (and companion animal owners) with patient-based research.

The navigation of ethical practice and delineation of acceptable and unacceptable animal use in the very different context of animal research laboratories has been referred to as "ethical boundary work" [38]. Veterinary professionals in

the present study also engaged in negotiating ethical boundaries, and described how these boundaries were coloured by the priorities of veterinary practice. In particular, participants discussed the tension between a veterinary professional's primary role to protect animal health and [17], the need to provide a reliable service to paying clients, and the practice of conducting research involving animal patients, in which subjects are put at some degree of risk or harm – even if those risks are relatively minor, to potentially benefit a species as a whole. These concepts could be seen to be ethically incongruous [16,39], particularly given that some areas of clinical veterinary research, such as critical care research, have been criticised for poor practices, poor reporting, and potentially replicating studies unnecessarily [40]. Tensions between owner autonomy and veterinary professional primary duty of care towards animals were also expressed by some participants in the present study [28,41]. Whilst primarily in favour of research involving client-owned companion animal patients, these "in-between" processes around negotiation of ethical boundaries led to participants describing relatively conservative views in relation to the specific research scenarios described in this study. Ongoing calls for an internationally agreed framework around regulation and ethical best practice in clinical veterinary research, considering the issues of autonomy and informed consent [16,41], appears to be timely and important.

Human healthcare research offers important career opportunities for nurses and associated professionals, who are ideally placed to communicate with patients, collect data and to coordinate or run research studies [42]. The career benefits and personal satisfaction of being involved in patient-based research within a veterinary clinical setting was discussed by registered veterinary nurse participants in the present study. This approach offers multiple benefits in terms of career progression and retention together with cost and time efficiencies compared to an approach involving only, or predominantly, veterinary surgeons [42]. Improved opportunities for veterinary nurses to be involved in research with relevant training and support could provide an additional way to expand current primary research generation.

The limitations of traditional gold standard randomised controlled trials (RCTs), which are highly regulated and can be prohibitively expensive and challenging to conduct, and value of primary evidence generated from well-designed, prospective observational studies utilising patient data are increasingly recognised within medical [43] and veterinary healthcare. Technological advancements in digital collection of patient data (human healthcare and veterinary) and use of 'big data' epidemiological approaches provide cost-effective ways of generating primary research evidence. This is particularly important in veterinary clinical settings where research funding remains extremely limited compared to human healthcare. Use of machine learning models provides cost and time efficiencies through interrogation of large volumes of unstructured, text-based veterinary electronic health records. These approaches are already contributing to generation of primary evidence from large veterinary companion animal patient populations [44,45] and is a rapidly expanding area of current veterinary clinical research. Nevertheless, RCTs, including pragmatic and newer adaptive and platform designs, remain the optimal way to obtain high-quality, unbiased primary evidence of efficacy of new therapies and treatment protocols in both human [46,47] and veterinary medicine [48]. Increased undergraduate and postgraduate education around best-practice in the design and conduct of interventional studies including RCTs would be beneficial. Improved knowledge of key study features including suitable study power and awareness of stopping rules, and improved education to address ethical and welfare concerns discussed by some participants in the present study including clinical equipoise, use of placebos and monitoring of patient safety, also appears to be required.

## Limitations

This study investigated the views of veterinary professionals working in UK companion animal (small animal and equine veterinary) clinical settings. Therefore, the findings of the present study may not reflect the views and experiences of those working in other veterinary sectors or in other countries. Consistent with many qualitative studies, these findings are based on the results of in-depth interviews with a small number of veterinary professionals. We focussed specifically on companion animal patient-based research as this is very different from research involving production animal veterinary patients. Participants may also have been biased towards those with an interest in research and EBVM, although we tried

to address this as best as possible through the recruitment process. However, our findings are broadly consistent with the findings from other research studies investigating EBVM in veterinary professionals and with qualitative studies that have investigated attitudes of medical and other healthcare professionals towards patient-based research. Additional qualitative research in this area is warranted including further in-depth study of veterinary professionals, companion animal owners and other stakeholders involved in companion animal healthcare. Finally, while we included veterinary professionals from a range of settings (including corporate and independent private practices and university clinics) we did not seek to explore "trends" in the different experiences of participants from those settings; this is an area for further investigation in a larger sample size using quantitative methods.

## Conclusions

Generation of high-quality primary evidence through engagement of veterinary professionals in patient-based research has important benefits for animals and their owners. This includes generation of improved evidence that veterinary professionals and companion animal owners can use during informed decision-making around options for veterinary care, including likely outcomes and associated costs. This is particularly relevant to the current Competition and Markets Authority review of UK veterinary services for household pets [23] in which concerns about the ability for pet owners to make informed decisions about veterinary care is part of ongoing investigations. Continued improvement in knowledge and use of EBVM principles by UK veterinary professionals was evident [9,49]. Application of EBVM was unquestioned as the goal for responsible, up to date veterinary professionals and as a means of reducing uncertainty by quantifying the effectiveness of veterinary treatments and other interventions. Participants unanimously agreed that the evidence base for veterinary medicine remains poor and the repeatedly identified need to generate the primary evidence through patient-based veterinary research [50] remains important. Participants reported willingness to participate in patient-based research and recognised the associated career opportunities and other benefits it provides. Lack of time, perceived poor research skills and confidence, limited resources and support and other structural issues within the veterinary profession were major barriers to engagement with research. Additional education, funding and capacity building is required, utilising approaches used in global healthcare settings and paediatric medical research. This includes increased awareness of best practice in patient-based research, improved research skills and communication training, development of improved resources and communities of support and cost-efficient utilisation of digital patient data, and increased use of research-trained registered veterinary nurses. Current ethical, legal and regulatory issues relating to veterinary clinical research also need to be addressed at a national and international level.

## Supporting information

**S1 Appendix. Semi structured interview guide.** Script used during interviews with veterinary professionals in the study. (DOCX)

## Acknowledgments

We thank the veterinary professionals who kindly agreed to participate in this study, and Dr Rebecca Smith for comments on an early version of the manuscript.

## Author contributions

**Conceptualization:** Elizabeth Perkins, Debra C. Archer.

**Formal analysis:** Tamzin Furtado, Elizabeth Perkins.

**Funding acquisition:** Elizabeth Perkins, Debra C. Archer.

**Investigation:** Tamzin Furtado, Elizabeth Perkins, Debra C. Archer.

**Methodology:** Tamzin Furtado, Elizabeth Perkins, Debra C. Archer.

**Project administration:** Debra C. Archer.

**Resources:** Debra C. Archer.

**Supervision:** Debra C. Archer.

**Visualization:** Elizabeth Perkins.

**Writing – original draft:** Tamzin Furtado, Debra C. Archer.

**Writing – review & editing:** Tamzin Furtado, Elizabeth Perkins, Debra C. Archer.

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
