## [Decision Letter · Decision Letter 0]

12 Feb 2025

PONE-D-24-57716Contributing to evidence-based veterinary medicine: a qualitative study of veterinary professionals’ views and experiences of client-owned companion animal researchPLOS ONE

Dear Dr. Archer,

Thank you for submitting your manuscript to PLOS ONE. After careful consideration, we feel that it has merit but does not fully meet PLOS ONE’s publication criteria as it currently stands. Therefore, we invite you to submit a revised version of the manuscript that addresses the points raised during the review process.

We look forward to receiving your revised manuscript.

Kind regards,

Cord M. Brundage, D.V.M., Ph.D.

Academic Editor

PLOS ONE

Journal Requirements:

Reviewers' comments:

Reviewer's Responses to Questions

**Comments to the Author**

1. Is the manuscript technically sound, and do the data support the conclusions?

Reviewer #1: Yes

Reviewer #2: Yes

Reviewer #3: Partly

2. Has the statistical analysis been performed appropriately and rigorously? 

Reviewer #1: N/A

Reviewer #2: Yes

Reviewer #3: No

3. Have the authors made all data underlying the findings in their manuscript fully available?

Reviewer #1: Yes

Reviewer #2: Yes

Reviewer #3: No

4. Is the manuscript presented in an intelligible fashion and written in standard English?

Reviewer #1: Yes

Reviewer #2: Yes

Reviewer #3: Yes

5. Review Comments to the Author

Reviewer #1: Thank you for sharing this insightful study on the challenges and opportunities in evidence-based veterinary medicine. Your findings provide valuable perspectives on the barriers veterinary professionals face in generating and using research evidence. I wanted to offer some constructive feedback on potential limitations that could be considered for future research:

1. Small Sample Size and Generalizability – With only 20 participants, the findings may not fully capture the diversity of perspectives across the veterinary profession. A larger sample or mixed-methods approach could enhance generalizability.

2. UK-Specific Context – Since the study is based on UK veterinary professionals, findings may not fully apply to countries with different regulatory frameworks, funding models, or research cultures. Expanding to an international context could provide comparative insights.

3. Potential Selection Bias – Participants who volunteered may have a greater interest in research, meaning the perspectives of those less engaged in evidence-based medicine might be underrepresented.

4. Self-Reported Data – As with any interview-based study, responses are influenced by participants’ perceptions and recall, which may introduce some bias. Complementing this with observational or quantitative data could strengthen the conclusions.

5. Lack of Owner Perspectives – Since animal/pet owners play a key role in research participation, their views on discussing studies, consenting to participation, and trust in veterinary research could provide a more comprehensive picture.

6. Variability by Practice Type – While participants came from different clinical settings, it is unclear whether experiences differ between independent practices, corporate groups, and referral centers, which may have distinct research constraints.

These considerations do not diminish the study’s contributions but could help refine future research directions. Thank you again for this important work—I look forward to seeing how it informs improvements in veterinary research and practice.

Reviewer #2: All sections of the manuscript are well structured, unambigious, and easy to comprehend. The research is unique and very essential in advancement in veterinary practice as it obtains in human medicine. Appropriate methods were used and standard ethical protocol was followed. The results are detailed and extensively related to relevant previous reports. The conclusions are in tandem with the aim of the study. The limitations of the study were clearly explained and would be good as guide for researchers interested in conducting similar studies.

Reviewer #3: The authors present an evaluation of a specific subset of individuals (Surgeons and Nurses) and their relationship and thought process about using evidence-based veterinary medicine. They attempt to use a data-driven approach to query and evaluation, often intermixing initial review of information with later questions and driven results. The interview-styled evaluation attempts to draw parallels between the “idea” of evidence-based medicine and the practicality of application given the various small sizes and often niche papers or industry-driven research.

However, this study has significant limitations that they acknowledge partially but ultimately make it difficult to utilize this paper to further science. The simple part is that much of this research (and similar conclusions) has been done in human medical space that can quickly draw parallels. The authors acknowledge that they utilize a small number of professionals. Still, they focused explicitly on surgeons and nurses (reflected by the surgeons given their working proximity), which does not necessarily reflect generalizability. There are also limited conclusions possible, given the interlaced nature of utilizing current results to change future questions and focus.

The results of the study also have a well-understood nature that limits its impact on veterinary medicine. It is not commonly reported as a direct result, but the “trust” or “reliance” on external companies for their evidence has been reported over decades.

The core of the paper, however, is how we relate EBVM to our staff, but the size and scope of the study needs to be significantly expanded to be able to draw conclusions. I also recommend evaluating clients as part of this, as with a cash-based business, the client drivers can often help speed up change within the industry.

The current paper does not offer enough novel and statistically significant results to warrant publication. Still, I do believe that with an expanded scope (including primary practitioners, clients, and other specialties), identifying key drivers (word cloud analysis is an effective tool for some of this), and offering essential considerations to help veterinarians drive evidence-based medicine will increase this article’s weight and importance.

Minor comments:

- Some inconsistent use of abbreviations (sometimes VPs vs veterinary professionals)

- Having a header of Results and a sub-header of results can be confusing to readers

- There are some very lengthy sentences, which makes understanding the paper difficult.

6. PLOS authors have the option to publish the peer review history of their article (what does this mean? ). If published, this will include your full peer review and any attached files.

**Do you want your identity to be public for this peer review?** For information about this choice, including consent withdrawal, please see our Privacy Policy .

Reviewer #1: **Yes: ** Dr Samrat Kumar

Reviewer #2: No

Reviewer #3: No

---

## [Author Response · Author response to Decision Letter 1]

25 Mar 2025

RESPONSE TO THE ACADEMIC EDITOR

Dear Dr. Brundage,

We thank you for your time and assistance with review of our submitted manuscript. We have addressed the journal requirements for the re-submission. A point-by-point response to the reviewers’ comments is in the second section.

Journal Requirements:

Here is our updated final disclosure statement: “This work was funded through the University of Liverpool Policy Support Fund awarded to DA through a competitive application round (2021-22). This fund is a Research England award to the University of Liverpool. The funders played no part in the study design, data collection and analysis, decision to publish, or preparation of the manuscript.”

1. Please ensure that your manuscript meets PLOS ONE's style requirements, including those for file naming. Done as requested and changes are highlighted within the text. Line 199 has been modified to ‘Interview results’ to avoid repetition of “results” as picked up by Reviewer 3.

2. Please include your full ethics statement in the ‘Methods’ section of your manuscript file. In your statement, please include the full name of the IRB or ethics committee who approved or waived your study, as well as whether or not you obtained informed written or verbal consent. If consent was waived for your study, please include this information in your statement as well. Done as requested – we assume that this relates to the fact that we had not included written or verbal consent in the ethics statement within the main body of the paper. This has been added to the manuscript.

3. We note that the grant information you provided in the ‘Funding Information’ and ‘Financial Disclosure’ sections do not match. When you resubmit, please ensure that you provide the correct grant numbers for the awards you received for your study in the ‘Funding Information’ section. Modified as requested.

4. We note that you have indicated that there are restrictions to data sharing for this study. For studies involving human research participant data or other sensitive data, we encourage authors to share de-identified or anonymized data. However, when data cannot be publicly shared for ethical reasons, we allow authors to make their data sets available upon request. Please update your Data Availability statement in the submission form accordingly. Done as requested; we cannot publicly share the data for ethical reasons due to data containing potentially identifying information about individuals.

5. Please include captions for your Supporting Information files at the end of your manuscript, and update any in-text citations to match accordingly. Done as requested

6. Please review your reference list to ensure that it is complete and correct. Done as requested; it has been correctly formatted, several typos have been corrected and the list is now complete and correct. We did not leave all the track changes in as there were so many.

Other changes made

We have shortened some sentences as requested by Reviewer 3 and have corrected a few typos (see track changes in the manuscript).

In Table 1, the scenarios have been relabelled 1, 2 or 3 (instead of A, B or C) as they are described in the text in the manuscript; we apologise for not correcting this in the original submitted version.

RESPONSES TO THE REVIEWERS

We thank the reviewers for their time evaluating our paper. Below are our responses to the comments made.

1. Is the manuscript technically sound, and do the data support the conclusions?

Reviewer #1: Yes

Reviewer #2: Yes

Reviewer #3: Partly We have addressed this comment in Section 5

2. Has the statistical analysis been performed appropriately and rigorously?

Reviewer #1: N/A

Reviewer #2: Yes

Reviewer #3: No We have addressed this comment in Section 5

3. Have the authors made all data underlying the findings in their manuscript fully available?

Reviewer #1: Yes

Reviewer #2: Yes

Reviewer #3: No We have addressed this comment in Section 5

4. Is the manuscript presented in an intelligible fashion and written in standard English?

Reviewer #1: Yes

Reviewer #2: Yes

Reviewer #3: Yes

5. Review Comments to the Author

Reviewer #1: Thank you for sharing this insightful study on the challenges and opportunities in evidence-based veterinary medicine. Your findings provide valuable perspectives on the barriers veterinary professionals face in generating and using research evidence. I wanted to offer some constructive feedback on potential limitations that could be considered for future research:

Authors: thank you for your comments which are appreciated.

1. Small Sample Size and Generalizability – With only 20 participants, the findings may not fully capture the diversity of perspectives across the veterinary profession. A larger sample or mixed-methods approach could enhance generalizability.

Thank you for these comments which we agree with, and which are discussed in the limitations section. Follow-on work will utilise a mixed methods and quantitative approach. However, we would defend the sample size used in this study; references to support this are listed below. This is an area in which minimal research has been conducted within the veterinary profession and the aim was to generate initial data from a small but broadly representative sample of veterinary professionals through an initial purely qualitative approach. As you will be aware, it is not the aim of qualitative research to generate generalisability, because it would be impossible to get both the breadth required to be generalisable and the depth required to capture each participants’ experience. It is therefore standard practice for qualitative studies to have a much smaller sample size than quantitative studies, but to aim for much more context and depth than could ever be captured in numbers. Instead of number, qualitative studies are determined by “information power”, often referred to in relation to the idea of “theoretical saturation”; that is, does the information provided by the number of participants so far, yield enough information to provide sufficient information on the research question? And, are the themes being found in later interviews now becoming familiar, enabling the researcher to posit that they have identified some key areas which explain the experiences of participants? This study used a combination of those methods, alongside the practical limitations drawn by the study’s timeline.

Braun, V., & Clarke, V. (2019). To saturate or not to saturate? Questioning data saturation as a useful concept for thematic analysis and sample-size rationales. Qualitative Research in Sport, Exercise and Health, 13(2), 201–216. https://doi.org/10.1080/2159676X.2019.1704846

Malterud, K., Siersma, V. D., & Guassora, A. D. (2016). Sample size in qualitative interview studies: guided by information power. Qualitative health research, 26(13), 1753-1760.

Sandelowski, M. (1995). Sample size in qualitative research. Research in nursing & health, 18(2), 179-183.

2. UK-Specific Context – Since the study is based on UK veterinary professionals, findings may not fully apply to countries with different regulatory frameworks, funding models, or research cultures. Expanding to an international context could provide comparative insights.

Thank you for your comments. Incorporating other regulatory frameworks or funding models would have been too broad (as discussed above) and was beyond the resources we had available to us. We agree this would be interesting for future studies and is an area of planned ongoing work. We consider that we have made this clear within the limitations section of the paper already (lines 750-753): “This study investigated the views of veterinary professionals working in UK companion animal (small animal and equine veterinary) clinical settings. Therefore, the findings of the present study may not reflect the views and experiences of those working in other veterinary sectors or in other countries.”

3. Potential Selection Bias – Participants who volunteered may have a greater interest in research, meaning the perspectives of those less engaged in evidence-based medicine might be underrepresented.

Yes, we agree with this statement; the potential for this type of bias is already stated in the limitations section of the paper (lines 756-761):” Participants may also have been biased towards those with an interest in research and EBVM, although we tried to address this as best as possible through the recruitment process. However, our findings are broadly consistent with the findings from other research studies investigating EBVM in veterinary professionals and with qualitative studies that have investigated attitudes of medical and other healthcare professionals towards patient-based research”

4. Self-Reported Data – As with any interview-based study, responses are influenced by participants’ perceptions and recall, which may introduce some bias. Complementing this with observational or quantitative data could strengthen the conclusions.

Thank you; agreed. This was beyond the remit of the current project and part of planned follow-on work. However, we have addressed this in the discussion and do not feel that this precludes this work being published in its current form.

5. Lack of Owner Perspectives – Since animal/pet owners play a key role in research participation, their views on discussing studies, consenting to participation, and trust in veterinary research could provide a more comprehensive picture.

Thank you – we agree with these comments. However, this was beyond the remit of the current project which specifically focussed on the veterinary perspective. The owner aspect is a study in itself and is an area of ongoing work by our group. This has already been commented on in the paper (line 763): “Additional qualitative research in this area is warranted including further in-depth study of veterinary professionals, companion animal owners and other stakeholders involved in companion animal healthcare.”

6. Variability by Practice Type – While participants came from different clinical settings, it is unclear whether experiences differ between independent practices, corporate groups, and referral centers, which may have distinct research constraints.

Thank you, and yes, we agree. This may play a part, but again was beyond the remit of this current project and would be important data to obtain in follow-on studies. We have added a sentence in the discussion based on these comments to make this clear (lines 768 – 771): “Finally, while we included veterinary professionals from a range of settings (corporate and independent private practice, University clinics) we did not seek to explore “trends” in the different experiences of participants from those settings; this is an area for further investigation in a larger sample size using quantitative methods.

These considerations do not diminish the study’s contributions but could help refine future research directions. Thank you again for this important work—I look forward to seeing how it informs improvements in veterinary research and practice.

Thank you for your kind and constructive comments, which are appreciated. We are looking forwards to addressing some of the issues you have correctly highlighted in our ongoing research in this area.

Reviewer #2: All sections of the manuscript are well structured, unambigious, and easy to comprehend. The research is unique and very essential in advancement in veterinary practice as it obtains in human medicine. Appropriate methods were used and standard ethical protocol was followed. The results are detailed and extensively related to relevant previous reports. The conclusions are in tandem with the aim of the study. The limitations of the study were clearly explained and would be good as guide for researchers interested in conducting similar studies.

Thank you for your positive and constructive comments and feedback, which is appreciated.

Reviewer #3:

We thank the reviewer for their time and comments. However, it would appear there are some fundamental misunderstandings around several key aspects of this study. Almost all of these comments do not align with the those of the other reviewers.

The authors present an evaluation of a specific subset of individuals (Surgeons and Nurses) and their relationship and thought process about using evidence-based veterinary medicine.

This is not a study of ‘Surgeons’ and ‘Nurses’. Veterinary surgeons is the standard term for a veterinarian in the UK (as per the Royal College of Veterinary Surgeons, UK) and does not mean that they are a “surgeon” in the context of human medicine. Our study incorporates different types of veterinary surgeons including first opinion / primary care (the same as a human general practitioner) and specialist (Boarded) veterinary surgeons with surgical, internal medicine etc expertise. This information is provided in Table 2 where primary care and referral practice are clearly stated.

However, we have added the term ‘veterinarians’ to line 111 and to the legend for Table 2 to make this clear to other readers who may also be unfamiliar with the term ‘veterinary surgeon’.

They attempt to use a data-driven approach to query and evaluation, often intermixing initial review of information with later questions and driven results. The interview-styled evaluation attempts to draw parallels between the “idea” of evidence-based medicine and the practicality of application given the various small sizes and often niche papers or industry-driven research.

The reviewer does not appear to be familiar with social science methodology. Unlike quantitative studies, where statistical significance is paramount, qualitative research is designed to explore nuanced perspectives and generate rich, contextualized understandings. As leading scholars in qualitative methods have demonstrated, small samples—especially when carefully selected for relevance and expertise—can yield highly valuable contributions to knowledge. There is a huge body of work utilising social science approaches in human healthcare research using the same methodology.

EBVM is perfectly achievable (and is indeed a RCVS professional requirement for veterinary surgeons) in the same way that EBM has been utilised in the healthcare field so we disagree with (what we understand to be) the comment around the practicality of application of evidence-based medicine as it is possible to undertake large scale, non-industry driven research.

However, this study has significant limitations that they acknowledge partially but ultimately make it difficult to utilize this paper to further science.

We disagree strongly with this statement. Our contribution lies in the conceptual and thematic insights we provide, which cannot be reduced to numerical significance testing. Our findings are novel in the veterinary field and will be vital to help improve application of EBVM in exactly the same way that this challenge has been approached in human healthcare, particularly in low and middle income settings, as stated in the paper.

The simple part is that much of this research (and similar conclusions) has been done in human medical space that can quickly draw parallels.

We take issue with this statement and strongly disagree with these comments. The assumption that previous human medical research negates the need for this study in a veterinary context is completely unfounded and frustrating.

Whilst parallels to findings from similar studies in human medical research exist, and have been referred to in the discussion, veterinary practice presents distinct challenges that require specific contextual investigation. Fundamental differences between medical and veterinary healthcare exist e.g. owners electing for voluntary euthanasia on economic grounds where affordable treatment options exist (we are certain that no medical healthcare professional will be required to deal with this request on a regular basis) or owners pursuing treatment which may not be

---

## [Editor Report · Decision Letter 1]

30 Mar 2025

Contributing to evidence-based veterinary medicine: a qualitative study of veterinary professionals’ views and experiences of client-owned companion animal research

PONE-D-24-57716R1

Dear Dr. Archer,

We’re pleased to inform you that your manuscript has been judged scientifically suitable for publication and will be formally accepted for publication once it meets all outstanding technical requirements.

Kind regards,

Cord M. Brundage, D.V.M., Ph.D.

Academic Editor

PLOS ONE

---

## [Editor Report · Acceptance letter]

PONE-D-24-57716R1

PLOS ONE

Dear Dr. Archer,

I'm pleased to inform you that your manuscript has been deemed suitable for publication in PLOS ONE. Congratulations! Your manuscript is now being handed over to our production team.

Kind regards,

on behalf of

Dr. Cord M. Brundage

Academic Editor

PLOS ONE